# ZAF, the first open source fully automated feeder for aquatic facilities

**Merlin Lange\*, AhmetCan Solak, Shruthi Vijay Kumar, Hirofumi Kobayashi, Bin Yang, Loïc Alain Royer\***

Chan Zuckerberg Biohub, San Francisco, United States

**Abstract** In the past few decades, aquatic animals have become popular model organisms in biology, spurring a growing need for establishing aquatic facilities. Zebrafish are widely studied and relatively easy to culture using commercial systems. However, a challenging aspect of maintaining aquatic facilities is animal feeding, which is both time- and resource-consuming. We have developed an open-source fully automatic daily feeding system, Zebrafish Automatic Feeder (ZAF). ZAF is reliable, provides a standardized amount of food to every tank, is cost-efficient and easy to build. The advanced version, ZAF+, allows for the precise control of food distribution as a function of fish density per tank, and has a user-friendly interface. Both ZAF and ZAF+ are adaptable to any laboratory environment and facilitate the implementation of aquatic colonies. Here, we provide all blueprints and instructions for building the mechanics, electronics, fluidics, as well as to setup the control software and its user-friendly graphical interface. Importantly, the design is modular and can be scaled to meet different user needs. Furthermore, our results show that ZAF and ZAF+ do not adversely affect zebrafish culture, enabling fully automatic feeding for any aquatic facility.

## Editor's evaluation

This is a nice example of an accessible tool for aquatic science, which will be valuable to an array of different researchers.

**\*For correspondence:**
merlin.lange@czbiohub.org (ML);
loic.royer@czbiohub.org (LAR)

## Introduction

The zebrafish (*Danio rerio*) is a well-established animal model in biology, with increasing use in different fields (*Kinth et al., 2013*; *Lidster et al., 2017*), including developmental biology (*Lawson and Wolfe, 2011*), neuroscience (*Wyatt et al., 2015*) and genetics (*Lieschke and Currie, 2007*). Among their advantages, zebrafish are vertebrates and have excellent optical properties as well as accessible genetics. Another essential feature of zebrafish is their low maintenance and husbandry cost (*Westerfield, 2000*). The development of commercial systems for zebrafish culture has helped advance zebrafish research (*Lawrence, 2007*). However, implementing a zebrafish facility remains a challenge for many small to medium sized laboratories due to cost and infrastructure issues. The most important aspect of zebrafish husbandry is the feeding, usually done manually at least two times a day by dedicated staff, using dry or living food like *Artemia nauplii* (*Lawrence, 2011*). Overall, manual feeding is not sufficiently accurate and can be time and resource prohibitive for labs without dedicated staff (*Candelier et al., 2019*). Very few technologies have been developed to automate zebrafish feeding and husbandry to help offset the challenges associated with implementing an aquatic facility. Some vendors propose fully automated solutions, but these are typically expensive, proprietary, incompatible with other systems, and require manual food filling before each feeding session. Other groups have recently published semi-automatic solutions that require human supervision (*Candelier et al., 2019*; *Tangara et al., 2019*). However, there is no open access and fully automated solution currently

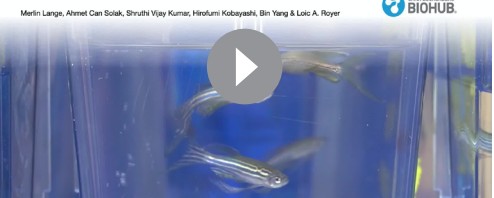

**Video 1.** ZAF presentation video, introducing the main features of the automatic feeders.

https://elifesciences.org/articles/74234/figures#video1

available. Ideally, all blueprints and instructions for building a flexible and scalable fully automated feeding system should be available to the zebrafish community. Importantly, such a design should be optimised for simplicity to facilitate adoption, avoiding complex 3D printing, mechanical assembly, or complex circuit board manufacturing. It should be easily assembled from inexpensive and commercially available parts, making it maximally accessible for non-experts. Here, we present ZAF (Zebrafish Automated Feeding) which satisfies all these requirements for automatic feeding of zebrafish as well as for any similar aquatic model organism.

## Results

### Automating aquatic husbandry

Establishing and maintaining an aquatic colony in research labs is not trivial. The colony requires a dedicated room with specific characteristics (e.g. temperature, water source, drain access, etc.) and regular monitoring by committed staff. To facilitate access to zebrafish research, and to reduce the amount of work needed to rear these animals, we developed a small semi-automated aquatic facility system that can be built within a regular wet lab. The only requirement being access to a sink and deionized water. To construct our facility, we used a stand-alone zebrafish rack, commercially available from different suppliers, that requires only minimum maintenance because these systems typically monitor water quality and automatically adjust water pH and conductivity. We then enclosed this system inside of a large indoor tent (*Figure 1—figure supplement 1*), and equipped this tent with a smart heating system to control the temperature, a carbon purifier to regulate humidity and odors, cameras for remote monitoring, and water sensor ropes to detect leaks. Once this basic life-support is provided the only missing feature to attain full automation is automatic feeding which is important to reduce staff workload (mainly during weekends and holidays) and standardize feeding. We introduce two affordable and easy to build automatic feeding systems: ZAF and ZAF+ (*Video 1*). Parts list, building instructions, and detailed blueprints to build your own ZAFs are provided in the *Supplementary file 1* and the latest version of this material can be found in the in the accompanying wiki (github.com/royerlab/ZAF/wiki). ZAFs are affordable and leveraging only commercially available parts (*Supplementary files 2 and 3*) We also provide the open-source python-based software to run the device, with command line interface (CLI) for ZAF and a stand-alone graphic user interface (GUI) for ZAF+.

### ZAF basic workflow

ZAF's design relies on mixing water with dry zebrafish food and then distributing this mix to all fish tanks. The basic operating principle of ZAF is simple: a servo motor rotates a food canister to dispense food into a container directly filled with water. This food-water mixture is then distributed to the tanks using pumps and a manifold tubing system. ZAF consists of three main modules: (i) electronics, (ii) tubing and pumps, and (iii) food preparation (*Figure 1a and b*). The electronics module is comprised of a credit card-sized computer (Raspberry Pi 3 B+) augmented with an extension board ('servo hat') that sends signals to various motor controllers to trigger pumping and valve opening (detailed description of the electronic circuit in *Figure 1—figure supplement 2* and more construction details in the *Supplementary file 1*). The Raspberry Pi 3 B + is connected to a touch screen and keyboard for easy user interfacing with the command-line interface. Several feeding programs can be added, modified and deleted. The amount of food delivered is constant across all tanks and can be modified by adjusting the food container opening as well as the degree of servo rotation. The tubing and pumps module is the central element in the food distribution system. The pumps mix food and water and distribute the mixture to the tanks. In ZAF, an air pump is used to stir and mix the food and water (*Figure 1a*). A splitter panel directs the liquid flow through the tubes leading to the individual

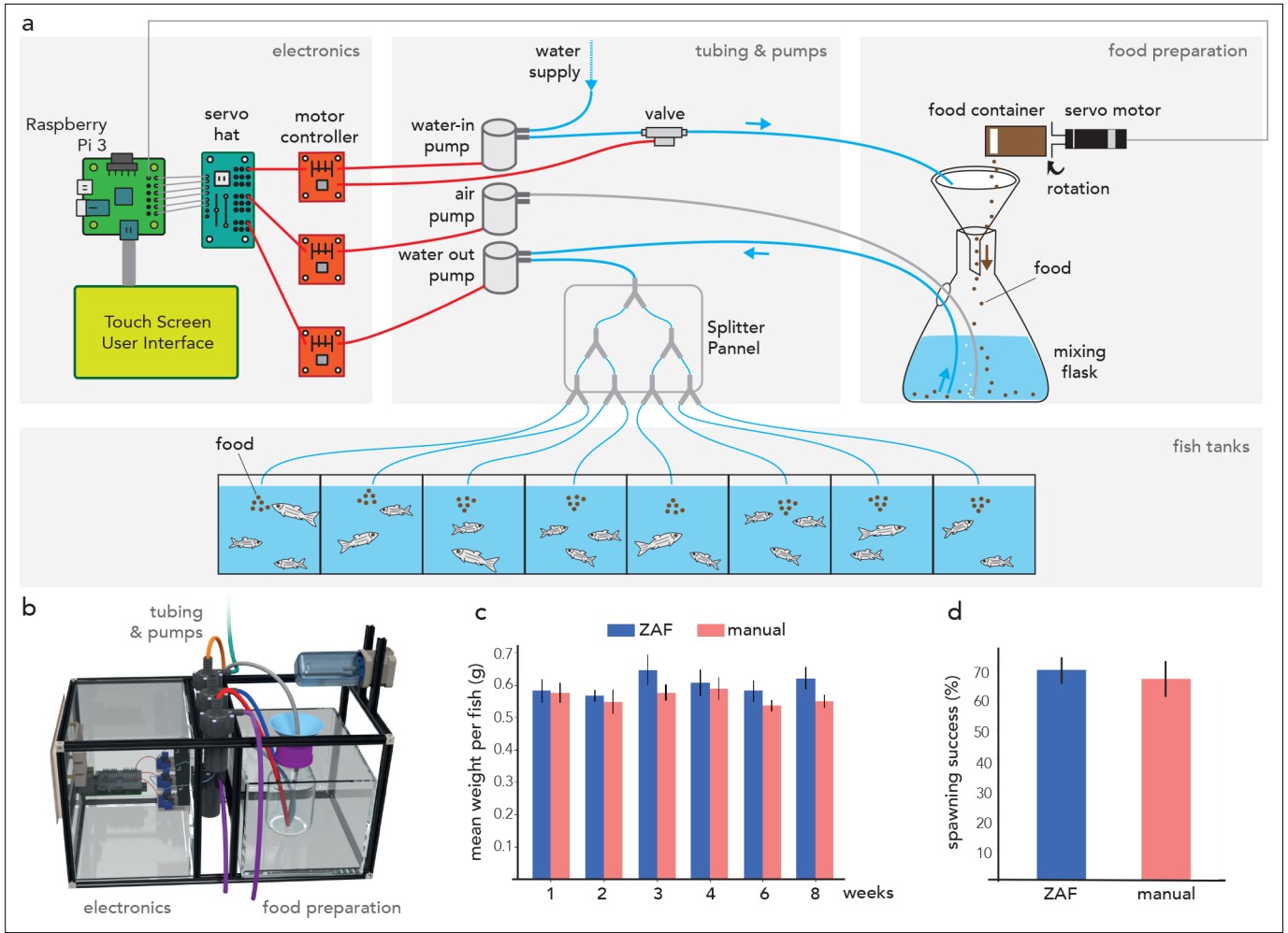

**Figure 1.** ZAF as a simple solution for aquatic facility feeding. (**a**) Schematic representation of ZAF's three main modules with their key components. Basic electronic wiring is also shown. ZAF is designed to distribute the same food quantity to all (fish) tanks. (**b**) 3D visualization of the different ZAF modules: electronics, tubing, pumps, and food preparation. (**c**) Variation of the fish mean weight over 8 weeks during ZAF feeding (n = 7) versus manually-fed fish (n = 6). (**d**) Spawning success for ZAF fed fish versus a manually-fed fish (spawning evaluated at weeks 2 and 6). All bars indicate s.e.m.

The online version of this article includes the following figure supplement(s) for figure 1:

**Figure supplement 1.** Automatising Zebrafish husbandry.

**Figure supplement 2.** ZAF electronic diagram.

**Figure supplement 3.** ZAF tubing.

**Figure supplement 4.** Water parameters log, pH and conductivity, during ZAF testing over a 3- month period.

tanks (*Figure 1—figure supplement 3*). A valve was added downstream of the water-in pump to prevent overflow or water leak in the device. Finally, for the food preparation module, we repurposed a commercially available aquatic food container and attached it to a servomotor for precise rotation control. For the mixing flask, we used a simple 200 ml plastic lab flask equipped with a funnel. To prevent water leaks, the food preparation container is placed in a water containment box. Additionally, we added a water sensor connected to a safety pump that, when activated, will remove any spilled water from this containment box. Once all parts have been delivered (see detailed list on *Supplementary file 2*), building ZAF in a few hours is feasible by following the instructions on our publicly available wiki. Both ZAF systems are highly modular and scalable: the number of tanks can be easily increased to meet the needs of larger aquatic facilities. For example, the system described in *Figure 1a* is designed for eight tanks but can be scaled up by adding extra pumps and by extending the splitter panels.

## ZAF structure and performance

The three modules that constitute ZAF are housed in a metal frame built with the versatile Makerbeam prototyping system. We provide all the detailed instructions for the hardware construction in the Supp. Information and in our wiki. The size of the automatic feeder can be adjusted from the baseline, which has a width of 15 inches, depth of 9 inches, and height of 9 inches (*Figure 1b*). Our prototype for the automatic feeding was sized for 16 zebrafish tanks. Distribution of complete nutrition dry food (Gemma-300 - Skretting Zebrafish) was calibrated according to the amounts recommended by the manufacturer. It was important to evaluate the impact of ZAF feeding versus manual feeding on fish health and fecundity. For this, we measured the weight of adult fish fed with the two techniques over 8 weeks and found no statistical difference (*Figure 1c*). Additionally, we found no excess mortality over the 8-week period for fish fed with the automatic device (zero fish died out of 92) versus manual feeding (one fish died out of 33). During the same period, we evaluated the fecundity of the fish and observed no difference between the two populations (*Figure 1d*). Additionally, the automatic feeding does not affect the water quality of our facility over a period of three months (*Figure 1—figure supplement 4*). Taken together, ZAF is appropriate for the feeding of a homogeneous fish population (i.e. tanks with a relatively equivalent number of animals) and it does not affect fish health nor fecundity.

## ZAF+ enables flexible, tank-specific feeding

While ZAF is an effective system for feeding multiple tanks with similar numbers of animals, it lacks precise control of food distribution to individual tanks. This can be problematic for aquatic facilities that have either disparate tank sizes or varying fish densities. To overcome this problem, ZAF+ was created to control food flow both spatially and temporally by adding valves upstream of each tank (*Figure 2a and b*). The ZAF+ software allows users to configure feeding parameters such as feeding frequency, timing, and quantity, as well as which tanks need feeding. With this system users can individually control and distribute a precise amount of food for each tank. For a more detailed explanation of ZAF+ feeding sequence compared to the simpler ZAF version see *Box 1* (*Box 1—figure 1*). ZAF+ was built by reusing several ZAF modules. However, most modules (i.e. electronics, tubing and valves, food preparation) were improved. We list all necessary components to build ZAF in the *Supplementary file 3*. Our design can be easily adapted to other needs by scaling up or down the various components. ZAF+ is larger (21" w x 12" d x 9" h) than the base ZAF version but still fits in a fish facility. To control the additional valves we added a micro-controller (Arduino Mega) for all pumps and valves, which permits limitless scalability by daisy-chaining multiple such controllers (*Figure 2—figure supplement 1*). The tubing and pumps module is extended to use a manifold to split the flow (*Figure 1—figure supplement 2*). Because of the more complex electronics and numerous wires in ZAF+, we enclosed all electrical components in a water-proof safety box. We used a touch screen for interfacing with the software, allowing the user to adjust settings such as the amount and timing of food delivery (Box 2—figure 1). ZAF+ can operate 7 days a week all year long, only requiring regular dry-food reloading as well as tube replacement. Tube replacement frequency varies on users usage and on facility environmental parameters (i.e. light and temperature). In our hands, we found that replacing tubing every 12 weeks takes one hour and is sufficient to keep tubes reasonably clean (*Figure 1—figure supplement 3*). We evaluated ZAF+ performance on both high- and low-density tanks, which is easily done through the user-friendly user interface. The fish were assessed for mean weight over 8 weeks. Overall, we observed no difference in the mean weights compared to the manually fed control group (*Figure 2c*). We then evaluated the reproduction of fish fed with ZAF+ and found no significant differences with fish fed manually. Finally, ZAF+ does not affect the water quality during a three months period (*Figure 1—figure supplement 4*) nor fish mortality. Thus, ZAF+ is a viable solution for full feeding automation in aquatic facilities.

## Discussion

In the present report, we introduced two aquatic feeding devices, ZAF (*Figure 1*) and ZAF+ (*Figure 2*) and evaluated their applicability for zebrafish feeding and husbandry over 8 weeks. ZAFs are effective and cheap solutions to overcome staffing issues on weekends and holidays and we hope will help disseminating aquatic animal models in research institutes. Interestingly, our devices can be easily

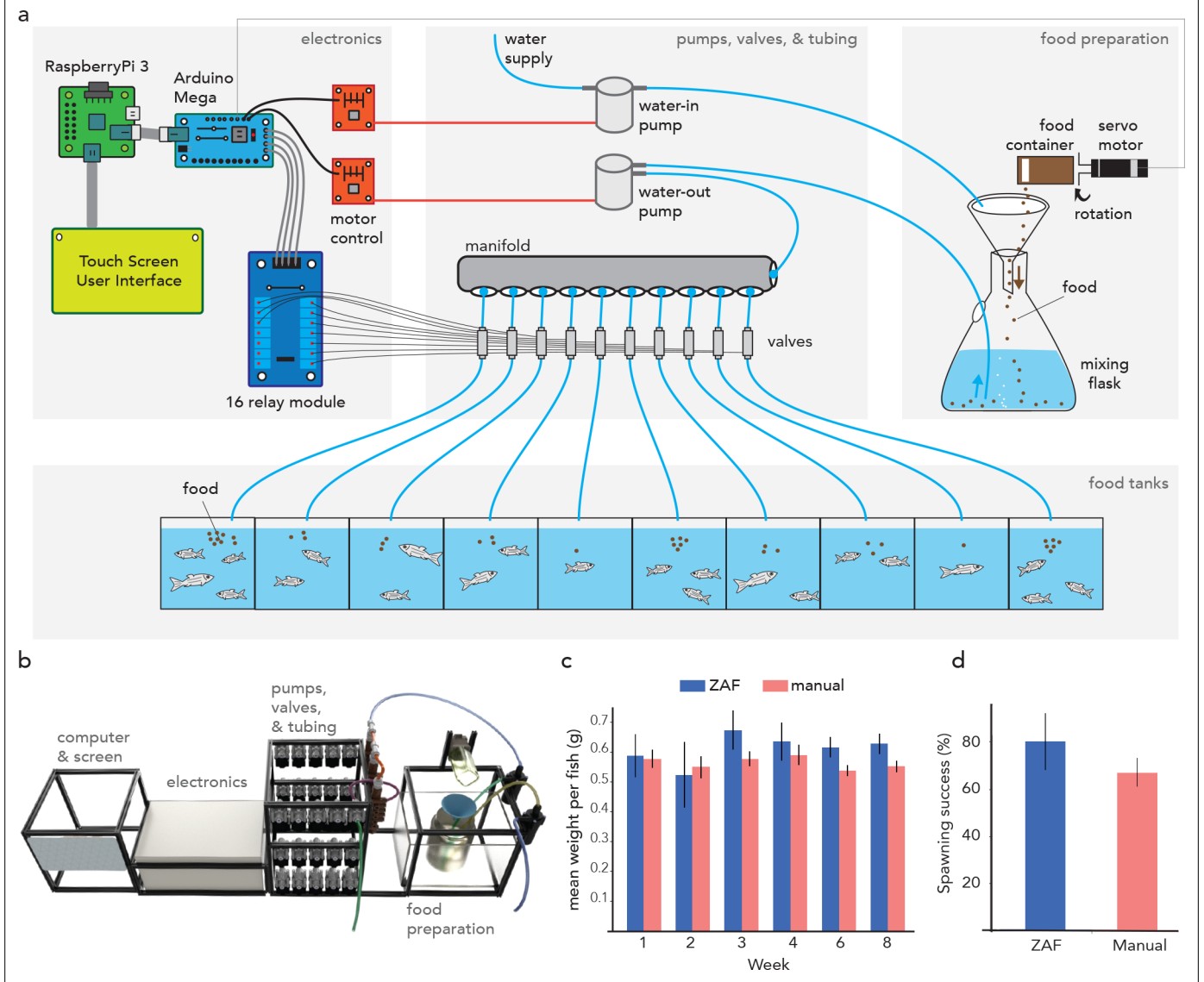

**Figure 2.** ZAF+ is an advanced version of ZAF that can modulate food delivery per tank depending on fish density. (**a**) Diagram of ZAF+. Electronics consist of: a Raspberry Pi 3, Arduino Mega, 16 relay module, an motor controllers. The water and food mix is pumped and sent via tubes to a manifold and valves that distribute it to specific tanks. (**b**) 3D representation of ZAF+ with extra space for the valves and a electronics box compared to the base ZAF version. (**c**) Evolution of the mean fish weight over 8 weeks of ZAF feeding (n = 6) versus the control group (n = 6). (**d**) Spawning success for fish fed by ZAF+ versus the control group. All bars indicate standard error mean.

The online version of this article includes the following figure supplement(s) for figure 2:

**Figure supplement 1.** ZAF+ electronics diagram.

**Figure supplement 2.** ZAF+ tubing.

**Figure supplement 3.** Tubing cleanliness evaluation.

**Figure supplement 4.** Water parameters log, pH and conductivity, during ZAF+ testing over a 3- month period.

adapted to behavioral studies (i.e. food conditioning) as well as any type of experimental design that requires pellets or liquid delivery, illustrating the versatility of our system. Both designs are fully open access (hardware and software), modular, scalable and highly adaptable. We also include instructions on installing a graphical user interface to run the automatic feeders. Both designs are relatively easy to build and do not require specialized training in electronics nor engineering. Importantly, ZAFs can easily be adapted to all commercially available aquatic facilities. ZAF+ is more robust than ZAF due to conceptual and technical improvements. While ZAF is easy to build, it does not offer control over

## Box 1. ZAF vs ZAF+ - the differences and how to choose the best for one's needs.

While ZAF distributes the same amount of food to all tanks, ZAF+ dispenses a variable quantity per tank as instructed by the user, typically based on fish density per tank. This advantage is counter-balanced by the higher sophistication of ZAF+ compared to ZAF. Both systems serve different needs which should be evaluated before construction. ZAF performs well for fish facilities with fish density variations across tanks of up to 30%. However, for higher density differences between tanks we strongly recommend ZAF+ instead which has several additional design upgrades such as stronger pumps and an electric safety box which increase reliability. The diagram on *Box 1—figure 1* illustrates the differences in running sequences between ZAF (left) and ZAF+ (right). Overall, they share many common features, like the quick distribution of food and water mix, to avoid pellets dissolution in water and loss of nutrients. While ZAF prepares and distributes food for all tanks equally, ZAF+ enables individual programming per tank. We added to the program a priming function to remove any air in the pump and flood the suction line before each program run. Finally at the end of each food distribution sequence we programmed a cleaning step to rinse the system (i.e. tubes, pumps, and valves) by flushing water and then air (illustrated by the boxes 'system cleaning' in the figure). In the case of ZAF+, there is an additional cleaning steps after each food distribution to individual tank ('cleaning' boxes in the figure). Importantly, for ZAF+ cleaning steps for all tubes and valves even those not actively used for feeding, are necessary to restrain algal and bacterial growth in the system.

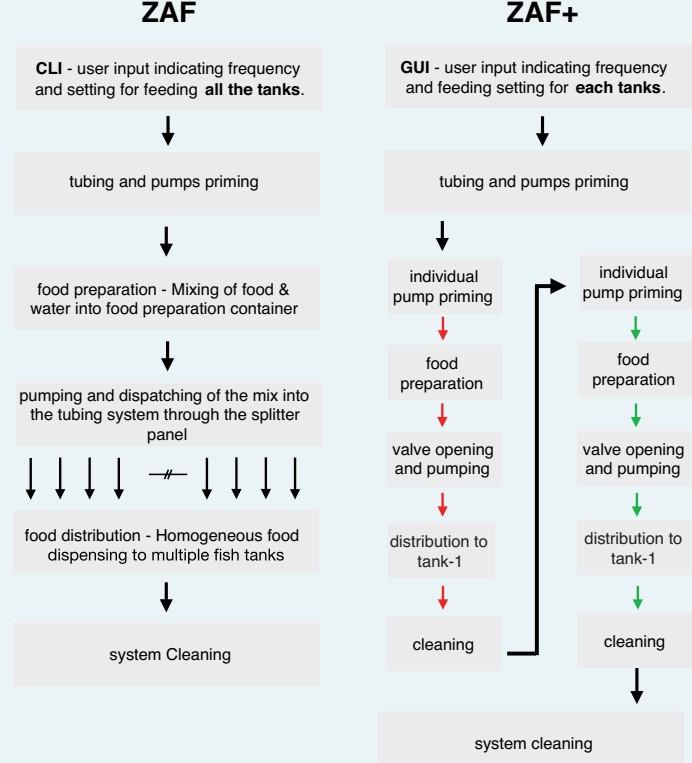

**Box 1—figure 1.** Diagram illustrating the differences during ZAF (left) and ZAF+ (right). ZAF is designed to distribute an homogeneous mix of food for all the tanks, whereas ZAF+ can control the amount of food per tanks.

## Box 2. A GUI for an efficient and simple ZAF+ control.

The guiding principle for the design of the control software and user interface for ZAF systems was simplicity and user friendliness. We hope that this will spur and facilitate adoption. The core control software for both devices is an open-source Python-based software running on a Raspberry Pi. All instructions for installation and operation can be found on our repository (github.com/royerlab/ZAF). The user interface contains three main tabs: (i) the 'dashboard' where users can select the running programs (*Box 2—figure 1* top), (ii) the 'log' panel that provides information on the currently running program, and (iii) the program panel which lets users change feeding parameters like scheduling (frequency, timing), food quantity, and the tanks to be fed (*Box 2—figure 1* bottom). Four levels of food quantity can be selected and calibration can be customized by changing the servo rotation value in the configuration file (see *Supplementary file 1*). Once a day, a special cleaning program flushes water and then air through the system. This program is analogous to a feeding program but without actual food distribution – this limits accumulation of algae and bacteria.

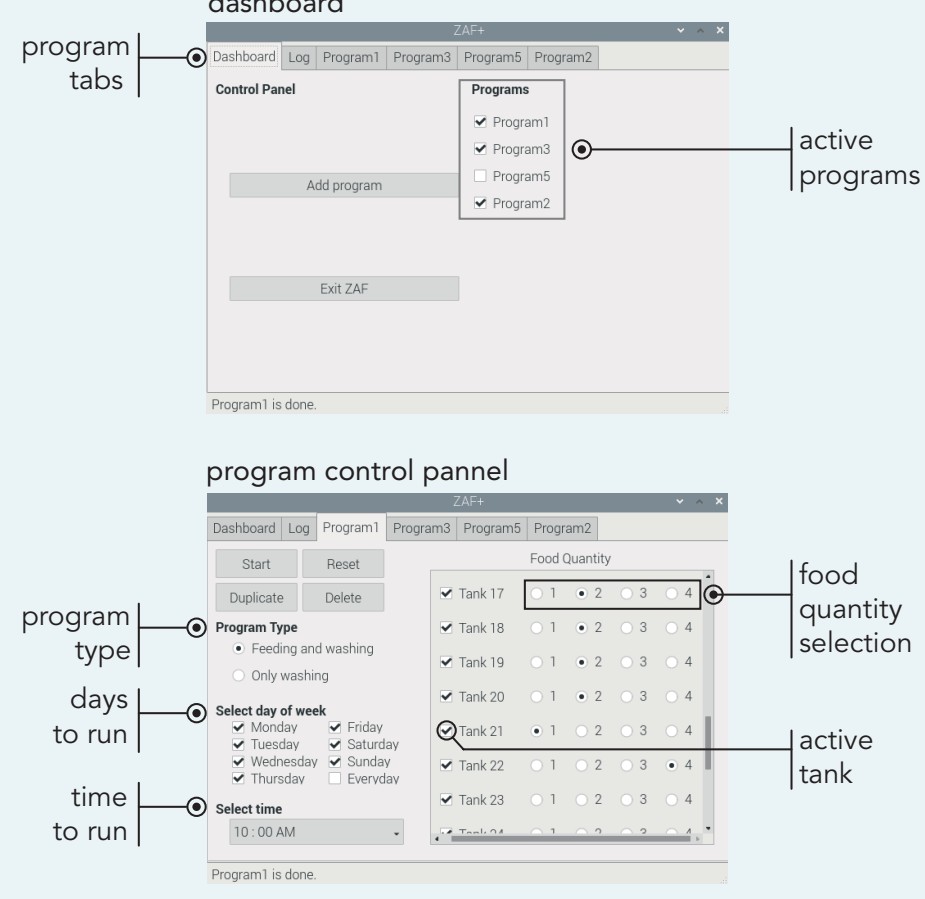

**Box 2—figure 1.** ZAF+ Graphic User Interface main panels. On the top panel 'Dashboard' displays the overview of the entire program. Bottom part illustrates the 'Program tab', where you set all the conditions for feeding and washing.

food distribution per tank. ZAF+ construction is more complex but is amenable to feeding variable numbers of fish per tank, or different tank sizes (*Box 2*). To maintain the tubes clean and unclogged after each feeding sequence, we programmed a cleaning step as described and illustrated in *Box 1* with water and then air. Additionally, we run a cleaning program (no food, water, and air only) to wash the system and prevent clogging (*Box 2*). Both devices can run 7 days a week all year long. For optimal

performance, we do recommend manual cleaning or replacing of the tubes every 10 weeks. We tested both ZAF systems in our fish facility for a total of 19 months (9 months for ZAF and 10 months for ZAF+) and never suffered from major malfunctions, nor observed adverse health effects to our fish. To help with potential malfunctions, we provide a troubleshooting guide to common minor issues we encountered while building or operating ZAFs (*Supplementary file 4*). We invite all ZAFs developers and users to report any issues on the ZAF Github repository (github.com/royerlab/ZAF/issues). Lab automation is likely to be increasingly critical to improve productivity, efficiency and research reproducibility (*Check Hayden, 2014*; *Boyd, 2002*; *Almada et al., 2019*). However, the field of animal husbandry has not yet made enough progress toward full automation and this holds particularly true for aquatic facilities, which have not been modernized for decades. For these reasons, and given constraints on personnel, we developed our own fully automatic fish feeding system. We used inexpensive hardware such as the micro-computer Raspberry-Pi (*Gay, 2014*), the Arduino micro-controller, and the Python programming language (Van *Van Rossum, 1995*). Making your own design is not easy and requires time and fine-tuning (*May, 2019*; *Blow, 2008*) therefore to help others in the community we decided to document in our wiki (github.com/royerlab/ZAF/wiki) all the steps we took in creating our stand-alone solution (see also the Supp. Information for a snapshot of the wiki). The wiki also offers solutions and advices on how to scale up ZAFs to different fish facility specifications. In this work we have used dry food (Gemma from Skretting Zebrafish) because of its complete nutritional profile and positive effect on fish health (*Lawrence, 2011*; *Lawrence et al., 2012*; *Barton et al., 2016*). Automated feeding is most easily performed with dry food; however, the food container can be adapted to live food if required (data not shown). For example, in the case of *Artemia nauplii*, a popular zebrafish diet, fresh live food is prepared every day. We tested ZAF and ZAF+ with *Artemia nauplii* and the feeding works well.

Another important aspect of aquatic husbandry is breeding of fish fry. While we did not directly test breeding, our automatic feeders can be easily adapted to deliver different types of food specific to different ages by adding several servos and food containers. Another solution could be to build two devices, one for adult and another one for small fry. With the advent of new aquatic model organisms with similar breeding requirements as zebrafish (*Lawrence, 2011*; *Lawrence et al., 2012*; *Barton et al., 2016*), it is conceivable that both ZAF systems could be adapted to others species. Finally, we hope that by releasing ZAF as an open access project we will empower a large community of users to build their own ZAFs, adapt them to their needs, help each other, and, perhaps, develop the next generation system.

## Materials and methods

### Animals and husbandry

This research was done under a protocol reviewed and approved by the institutional animal care and use committee (IACUC) of University of California San Francisco (UCSF). The fish were kept in a stand-alone aquatic system (Techniplast, Italy) with water maintained at 28° and a diurnal cycle of 10 hr of dark and 14 hr of light (*Aleström et al., 2020*). The study was conducted on the wild-type EKW strain, casper mutant (*White et al., 2008*) and *h2afva:h2afva-mCherry* transgenic line (*Knopf et al., 2011*) (gift from Jan Huisken, Morgridge Institute for Research, Madison, USA). We only housed and used fish between 4 months and 18 months old. Manual feeding is done once a day, at the same time of the day, according to the manufacturer recommendations and fish density.

### ZAFs equipment

ZAFs are designed to be built with only commercially-available parts. The *Supplementary files 2 and 3* list the necessary parts used to build ZAF, and ZAF+, respectively. Most of the parts used are generic and can be replaced by similar parts with similar specifications. The only component that cannot be easily exchanged is the Raspberry PI computer, but this is not an issue as these are very easily sourced.

### ZAFs construction manual

In the *Supplementary file 1* we provide detailed instructions on how to easily build the system with tools present in most labs and easy to source components (github.com/royerlab/ZAF/wiki). There are also instructions on how to run the software and operate the graphical user interface. To build the

ZAFs frame we use the versatile and easy to use Makerbeam consruction system. For both ZAFs we use two different tubing sizes, for the pump tubing we use 3/8" outside diameter tubes, for the valves tubing we use 1/4" outside diameter tubes. We use either silicone or PVC based tubing because they have good specifications and are safe for food delivery (PVC based are more cost effective).

### ZAF electronics
The electronic core of ZAF is based on (i) A Raspberry a credit card size computer, (ii) A Servo Hat Board to drive Pulse Width Modulation outputs, like the pumps and valve, (iii) Motor Controller to control the DC motors (pumps and valve). All the pumps and valves connected to the motor drivers are plugged on a 12V and 10A power supply converter. The Raspberry Pi, the servo hat and all the electronic connected to the servo hat are running with 5V through the Raspberry Pi power.

### ZAF+ electronics
ZAF+ electronics are comprised of four different components i. A Raspberry Pi 3 B + to run the software and control the electronics, ii. two Arduino Megas Arduino 2,560 microcontrollers for the digital devices, iii. several motor controllers to control the various pumps, iv. 16 Relay Module interface board to drive current and control the valves. The two Arduinos are daisy-chained via a serial connection (the whole design can be extended by daisy-chaining more arduinos). A 12V power supply provides power to the electronics, except for the Raspberry Pi and the two Arduino Megas powered by the Raspberry Pi 5V.

### Code availability
The control software for both ZAFs as well as the corresponding graphical user interfaces are available as open-source code. We also provide instructions and a step-by-step guide on how to run the software (github.com/royerlab/ZAF/wiki/Software).

### Automatic feeding calibration - ZAF
We use the Gemma micro 300 (Skretting Zebrafish) food diet. Feeding is calibrated so that ZAF distributes 5% of the fish body weight per feeding. This follows the producers' recommendations. We feed the fish in our facility twice a day. Based on the number of fish, we calibrate the automatic device to distribute 5 g of food homogeneously to all tanks per run. This calibration is done by manually by adjusting the food container opening, and the amount of servo rotation.

### Automatic feeding calibration - ZAF+
Similarly to ZAF, we use the Gemma micro 300 (Skretting Zebrafish) diet and run the program twice a day. Food distribution is done per tank according to a 'food quantity selection' parameter that can be set on the user interface: '1' for low fish density, to '4' for high fish densities. Calibration is done in same way as for ZAF. The amount of food distributed per fish density is detailed in *Box 2*. The approximate amount of food required for different fish densities is as follows: Very low - 100 mg for to 2–4 fish, low - 200 mg for 5–8 fish, medium - 350 mg for 9–14 fish, large - 500 mg for 15 up to 20 fish.

### Fish weight and spawning measurements
To weigh the fish, we first took a clean petri dish and tared it on a weighing scale. Each fish was then dabbed on a tissue paper to remove excess water and then placed in the petri dish to weigh it. This was repeated for all fish individually. To demonstrate the feeding efficiency of ZAF+, we documented the weight of the fish over a period of 7 weeks. The fish were weighed every Monday from week 1 to week 8. Since ZAF+ has the potential to customize the amount of food given per tank based on the number of fish present, we chose two tanks - one with over 12 fish and the second with only four fish to ensure each tank receives the designated amount of food. These two tanks were kept with the same fish population during the whole evaluation. We used tanks from different rows (top and bottom) to verify that tube layout and length do not affect the feeding quality nor quantity. Similarly we tracked the breeding of the fish over 2 weeks. Three random fish were selected and two to three crosses were bred for each of them. Next day, we documented the number of crosses which bred for each of the lines and calculated the average of all the positive crosses.

## Acknowledgements

We thank the Chan Zuckerberg Biohub facility team, particularly Jennifer Mann, for their help and support to implement the fish facility and Rafael Gomez and Robert Pucinelli from the Chan Zuckerberg Biohub Bioengineering platform for advice. We thank Ashley Lakoduk and Mirella Bucci for critical comments and diligent proofreading of the manuscript. We thank Sandra Schmid for generous mentoring, proof-reading and advice. Funding for this work was provided by the Chan Zuckerberg Biohub.

## Additional information

### Competing interests

Merlin Lange, AhmetCan Solak, Loïc Alain Royer: A patent application has been filed covering the reported feeders (number 63/162,299). The other authors declare that no competing interests exist.

### Funding

| Funder | Grant reference number | Author |
| --- | --- | --- |
| Chan Zuckerberg Biohub | | Loïc Alain Royer |

The funders had no role in study design, data collection and interpretation, or the decision to submit the work for publication.

### Author contributions

Merlin Lange, Conceptualization, Data curation, Formal analysis, Investigation, Methodology, Writing – original draft, Writing – review and editing; AhmetCan Solak, Hirofumi Kobayashi, Software; Shruthi Vijay Kumar, Investigation, Resources; Bin Yang, Visualization; Loïc Alain Royer, Conceptualization, Funding acquisition, Investigation, Methodology, Project administration, Software, Supervision, Visualization, Writing – review and editing

### Author ORCIDs

Merlin Lange ⓘ http://orcid.org/0000-0003-0534-4374
Loïc Alain Royer ⓘ http://orcid.org/0000-0002-9991-9724

### Ethics

This research was done under a protocol reviewed and approved by the institutional animal care and use committee (IACUC) of University of California San Francisco (UCSF).

### Decision letter and Author response

Decision letter https://doi.org/10.7554/eLife.74234.sa1
Author response https://doi.org/10.7554/eLife.74234.sa2

## Additional files

### Supplementary files

- Supplementary file 1. Detailed construction instruction for ZAF and ZAF+.

- Supplementary file 2. ZAF+ parts list. The table lists the necessary parts to build ZAF. Most of the parts are generic and can be replaced by components with similar specifications.

- Supplementary file 3. ZAF+ parts list. The table lists the necessary parts to build ZAF+. Most of the parts are generic and can be replaced by components with similar specifications.

- Supplementary file 4. Troubleshooting guide. This table provides solutions to common minor issues encountered during ZAFs construction and operations.

- Transparent reporting form

## Data availability

We provide all instructions to build the hardware and all code for the software in the wiki: github.com/royerlab/ZAF.

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
