## [Editor Report]

This is a nice example of an accessible tool for aquatic science, which will be valuable to an array of different researchers.

---

## [Decision Letter]

[Editors' note: this paper was reviewed by Review Commons.]

**Decision letter after peer review:**

Thank you for submitting your article "ZAF – The First Open Source Fully Automated Feeder for Aquatic Facilities" for consideration by *eLife*. Your article has been reviewed by 3 peer reviewers at Review Commons, and the evaluation at *eLife* has been overseen by a Reviewing Editor and Richard White as the Senior Editor.

Based on the previous reviews and the revisions, the manuscript has been improved but there are some remaining issues that need to be addressed, as outlined below:

This is a nice paper and a great start. However, key information should be fully contained in the written document and not shifted to the wiki. For example, the component list etc should be found in a revised paper, not as a link to a wiki. The authors should carefully check that the manuscript is self-contained for all key points.

---

## [Author Response]

This is a nice paper and a great start. However, key information should be fully contained in the written document and not shifted to the wiki. For example, the component list etc should be found in a revised paper, not as a link to a wiki. The authors should carefully check that the manuscript is self-contained for all key points.

First, we decided to add to the manuscript two supplementary tables (Supp File 1 and 2), listing the necessary components necessary for the ZAFs, as you suggested.

Additionally, we added 4 supplementary figures to detail some key elements of ZAFs construction. The figure 1-S2 and Figure 2-S1 are the detailed blueprints for the electronic circuits of ZAF and ZAF+ respectively. Together with the detailed legends, the two figures will help the readers understand and possibly build the electronics of ZAFs, a central and complex element in the design of the automatic feeders. We also prepared the figure 1-S3 and Figure 2-S2, to describe the tubing and the connections across the different elements for ZAF and ZAF+. We added in the text reference to the new supplementary tables and figures to ameliorate the clarity of the manuscript and make sure the paper is now self-sufficient. This is in addition to a complete snapshot of the wiki in PDF form.

To explain more clearly the key instructions, we detailed critical aspects of the construction in the Methods section. Both, ZAF and ZAF+, have now an individual paragraph describing more in detail the electronics. This will help readers understand the central logic behind our design directly in the manuscript. There is also, in the revised version of the paper, a “ZAF Materials'” paragraph in the Methods section to clarify the fact that most parts are generic and easily replaced by parts with similar specifications. We added a few sentences in the Methods section “ZAF construction manual” to explain important points about the hardware construction.

Finally, we decided to add the troubleshooting table from the wiki (Supp File 3) to present solutions to the most common issues encountered during ZAFs construction and operations. We referred to the table in the Discussion section.

[Editors' note: we include below the reviews that the authors received from Review Commons, along with the authors’ responses.]

Reviewer #1 (Evidence, reproducibility and clarity Required):Lange et al. have developed an automatic feeding system for zebrafish facilities. The system is open-source and relatively easy to implement. The authors propose to systems, one that delivers the same amount of food for each aquarium (ZAF) and a second (ZAF+) that can adjust the amount of delivered food to each aquarium. The authors show no difference in fish weight, spawning and water quality, when fed using the automatic system or manually.In my opinion, the ZAF and ZAF+ are an excellent first approach to solve the complex problem of automatizing feeding in fish facilities. So far, only one company offers this option which is extremely expensive and demands a lot of maintenance.The manuscript is very well written and easy to follow. The supplementary material is very well detailed. It is clear that the authors intended to facilitate the implementation of the ZAF by potential users.

We appreciate the supportive comments from Reviewer 1 and address all comments below:

I just have a few comments regarding the system:1. The authors do not indicate how the system is cleaned. the system drains itself, but will any deposits of food remain in the tubes? Why is the system not flushed with clear water after each feeding? do the tubes get clogged?

We agree that the cleaning process was not clearly explained in the manuscript. We added clear sentences in ‘Box 1’ to describe the first cleaning step (see text and figure). Indeed, after each feeding, we flush water and then air into the tubes. Moreover, we explain in ‘Box 2’ that we have a second level of cleaning in the form of a special cleaning program that is run at least once a day with no food distribution (i.e. same program as used for feeding but without actual food mixed, we flush lots of clean water and then air in the system). Finally, in the discussion we clarify the different cleaning steps by adding extra explanations in the first paragraph. All these procedures and programs are very effective in preventing system clogging and in reducing the accumulation of debris and algae. After more than 19 months of ZAF and ZAF+ feeding in our facility we never experienced any tube clogging.

2. How long the system was tested for?

ZAF has run in the facility for 9 months and ZAF+ for 10 months since September. We added a sentence about the testing time in the discussion. We never experienced any major problems, only a few minor malfunctions, reported in the new troubleshooting table added to the wiki (suggested by the reviewer 2).

3. The ZAFs were used to feed 16 aquariums. For such a small rack, manually feeding takes less than 5 min. The authors should highlight that, at least for such small systems, the ZAFs will be especially very useful for feeding during weekends and holidays. Still, adding 16 commercially available small automatic feeders to each aquarium, could be simpler to implement.

As noticed by the reviewer, ZAFs are very useful when staff are not present (weekend, vacation, etc..). To emphasize on this particular point, we added a sentence in the discussion's first paragraph. The small automatic feeders available commercially are usually very difficult to attach to zebrafish facilities. Indeed, they can’t adapt to conventional lab aquatic facility racks because they are designed for pet aquariums. They also have less features compared to the ZAFs (difficult to adapt the food quantity, more food waste, cumbersome…). Additionally, by multiplying the number of devices (you need one small feeder per tank), one increases the risk of possible malfunction as well as the maintenance time required for food filling, cleaning etc… Thus, usage of small automatic feeders in laboratory aquatic housing racks is complex to adapt, a source of feeding error, is more cumbersome, and potentially more time consuming etc… They are simply not designed for professional aquaculture systems. Whereas ZAFs can be easily adapted to all the commercially available aquatic facilities. The fact that ZAFs simply ‘interface’ via tubes to fish facility racks makes them very versatile and unintrusive.

4. How do authors envisage implementing the ZAFs in much larger facilities (from 100 to 1000 tanks) ? Implementing a specific ZAF for each rack containing ~20 tanks may not be realistic.

Indeed, building multiple ZAFs will be complex and resource consuming. Thus, we designed ZAFs to be adaptable and modular, so one ZAF (or ZAF+) can easily be scaled to handle bigger facilities. The supplementary information and the wiki describe all the steps required to build a ZAF for 16 tanks and a ZAF+ for 30 tanks and many tips to scale up these devices without major modifications (up to 80 tanks for ZAF no restrictions for ZAF+). Of course, we do think that for truly large facilities, there is probably a sweet spot that balances the number of individual devices and the per-device capability. Having a single device feeding 1000 tanks is probably not wise, perhaps 5 devices for 200 tanks each (ZAF+) would be the best. Please note that the hardware cost and complexity scales roughly linearly with the number of tanks, no surprises here. Moreover, in the case of ZAF+ it is possible to use splitters to feed even more tanks from the same line (ZAF+).

We added pages in the ZAF/ZAF+ wiki, to help the users extend the feeding capacities of their desired ZAFs (see in the wiki “tips to scale up ZAF” – “tips to scale up ZAF+”). We also mentioned in the discussion the possibility of distributing food to more tanks with one device by increasing the outputs and referenced the wiki accordingly.

Having said this, we did not primarily design ZAFs for super large fish facilities, instead we designed the ZAF systems to facilitate adoption of fish models by many small and medium sized labs. We hope that our system will lower the bar for labs with moderate resources to get started with aquatic models, or labs that just want to ‘try’ a new aquatic model organism ‘on-the-side’.

5. How the length of the tubes influences the efficiency of feeding?

For ZAF the size of the tubes is very important because its design assumes homogeneous food distribution. In contrast, ZAF+ distributes the entire amount of water and food mix to each tank sequentially, so the tube length is not an issue. To make sure that tube length or tube layout is not affecting feeding efficiency we evaluated the weight of fish coming from tanks housed on two different rows (top and bottom). This was not clearly explained in the methods section – we changed the text to reflect that. Additionally, at the end of each ZAF+ run, the washing sequence runs a relatively large quantity of water to ensure that all food gets flushed out to the right tanks. We did not evaluate the precise amount of food delivered. However, after each feeding and cleaning all tubes are empty (see last sentences of the Box 2).

For feeding many tanks with the same ZAF it is necessary that the tubes will be of the same length. In that case, the system will become very cumbersome.

This is a fair concern. However, with a good design and with the help of cable tie it is very easy to organise the tubing and avoid ‘tube-hell’. We added a sentence to clarify the organisation in the wiki (see ZAF>Hardware>Tubing in wiki).

Longer tubes will probably need stronger pumps. What's the maximal length of tubes tested? That will limit the number of aquariums a ZAF can feed.

We never precisely measured that because the generic pumps we use are very powerful and their running time can be adjusted in the software by changing the constants in the code source (see troubleshooting new supplementary table). Therefore, the length of tubes should not be a limiting factor. Even stronger pumps (more amps) can be readily sourced on Amazon if really needed – although we doubt that this is necessary. Regarding the number of tanks that ZAF can feed, we simply recommend adding more pumps to increase its capacity (see previous comments or “tips to scale up ZAF” in the wiki).

Despite these comments, this is an excellent first approach, and the fact that the authors made it open-source and open access, make the ZAFs a very important contribution to the community. I have no doubt that some fish facilities will implement it and the community will help to improve it.

Thank you. We do think that the main benefit of an open source project is the community around it. We are currently collecting a growing list of interested labs and we are interested in organising an online workshop to discuss ZAF and ZAF+, with some talks, QAs, and more to help people getting started.

Reviewer #1 (Significance (Required)):This is the first open-source open-access automatic feeding system ever published. It is the first but very important step to the automation of research fish facilities.Referee Cross-commentingI agree with all the other reviewers.We also have to take into account that the system is a first prototype and although not ideal, it is open source. This will allow other labs to develop and improve their own models based on the ZAF.Reviewer #2 (Evidence, reproducibility and clarity (Required)):SummaryThe manuscript proposes an open source automated feeder for zebrafish facilities, although it would be amenable to other species. Overall, the manuscript is clearly written and easy to understand, the wiki is well sourced and clear. The commitment to open source is commendable.I have some questions regarding the long-term sustainability of this setup, as well as some discrepancies in the methods. Finally, as this aims to be useful to people with no engineering/electronics competence, I feel that it is not yet at a level that is accessible enough.

We are very pleased to see that the Reviewer appreciates our manuscript and our commitment to open access. We thank the Reviewer for his comments, in particular the comments about accessibility, and address them bellow:

Major commentsIt would be useful to have a centralized list of parts and components, which would make it easier for users to order all that is needed to assemble the ZAF or ZAF+, at the moment the information is distributed through the wiki as hyperlinks.

Extremely important! This was clearly an oversight from our part. We agree that a table listing all the components would help for constructing ZAF and ZAF+. We have added two tables in the wiki, one for ZAF and another for ZAF+, with all the necessary parts and components required to build both devices, with articles number, supplier and cost in dollars. Thanks to the reviewer for this excellent suggestion.

A troubleshooting guide for the common problems the team ran into (if any) would be useful for newcomers, even just as issues on the GitHub. The team may also consider some form of chat/forum/google group to allow discussions between users and experts.

The reviewer raised an important point, so we added to the ZAF wiki a troubleshooting guide to help users by listing the minor malfunctions that we observed. Additionally, users will be able to ask questions or report bugs on the ZAF GitHub using issues. Github issues will allow discussion and to track ideas and feedback within the ZAF user community. Finally, we just created a Gitter room: https://gitter.im/ZAF-Zebrafish-Automatic-Feeder to enable more interactive discussion.

Did the author observe any algal or bacterial growth in the feeding tubes over the 60 days? Do they have an estimate on how long the tubes stay "clean" enough? The authors mention tube changing every 10 weeks, can they explain the rationale, and did they assess the bacterial/algal contamination over that time? Do the splitter panel and food mixing flask also need replacing regularly?

After several weeks of usage, we indeed observed algal and bacterial growth in the tubes. In order to report and justify the need to change the tubes, we made a new supplementary figure illustrating the tube cleanliness over time, mainly algal and bacterial (see Suppl. Figure 3). We realized that 12 weeks is actually the optimal tubing renewing period in our facility. Algal and bacterial growth depends on the facility environment characteristics such as light intensity, water and air temperature, as well as feeding frequency and therefore might be adapted to the user’s facility specs. The splitter tubing can be changed based on user observations; we now mention this in the ZAF tubing supplementary material and on the wiki.

The authors mention that the tubing needs to be of similar length to ensure similar resistance and food distribution, did they compare the body weight of fish in racks at the top or at the bottom of their system? There are no overall differences, but maybe the bottom racks would receive slightly more food? Furthermore, did they quantify the differences in food/water delivery as a function of length differences?

The requirement for similar length is only necessary for ZAF because its accessible design assumes homogeneous distribution of the water-food mix through a passive splitter system which is susceptible to variable fluid resistance. In contrast, ZAF+ distributes the waterfood mix one tank at a time – ensuring that the correct amount of food is entirely flushed through any required tube length (the pumps are strong enough and we flush enough water). In the eventuality that the tube length is too long the user can adjust the pump running time by changing constants in the code (see troubleshooting table in the wiki and corresponding links). We thank the reviewer for suggesting evaluating the fish weight on fish from two extremal heights. Although we did not explicitly report this in the first version of the manuscript, we had actually anticipated this potential issue and therefore we did collect data for ZAF and ZAF+ for tanks housed on the top and bottom rows. We added a clear description of the weighting process in the material and method, highlighting the housing condition of the tanks tested. Finally, after each feeding run the tubes have been fully flushed and are empty without food debris or pellets remaining, irrespective of their sizes. So, we did not find it relevant to evaluate the precise amount of food effectively delivered as we control that already upstream.

Methods fish weight: The methods mention different amounts of food than the wiki, the rationale in the wiki is also different from the 5% of body weight outlined in the methods (which then matches the food amount of the methods). Which is the correct amount?

We thank the reviewer for noticing the inconsistency. The method numbers are the correct one, so we changed the wiki, we made a mistake when editing the figures. We wrote some sections of the wiki early during the development of the hardware. We unfortunately forgot to correct the inconsistencies.

The code is decently commented for scientific software with clear variable names, but I wonder how flexible it is if users cannot get access to the specific hardware (especially the pumps) used in ZAF/ZAF+? Can the authors briefly comment on this point?

The pumps are just built from 12V motors, you can find a large variety of such pumps online (Amazon, etc.) we have ourselves tried several, but there is no need to have the exact same model. We added a note to the tubing section of the ZAF and ZAF+ about that. The only components that cannot be easily exchanged are the Arduino and Raspberry PI, but that is not an issue as these are very easily sourced components.

The wiki could use more pictures or, to borrow the Proust Madeleine allusion, schematics akin to LEGO with more intermediary steps clearly outlined. Some pictures are also a bit small/busy (such as 2D and 2E in the frame section, or the magnet pictures), they may benefit from cartoons/schematics to clarify what is done. Alternatively, videos/time-lapses may help with better visualizing the assembly.

We appreciate the reviewer comments and added new pictures, schematic and extra legends in the wiki to help potential ZAFs builders. In the wiki for ZAF hardware we increased the size of all the pictures for all the different steps and added new legends to clarify the assembly. There are also now more pictures illustrating the construction steps (i.e. in “frame”, “pumps and valve”) and we added a simple schematic for “servo and food container”. Picture sizes have been increased in “ZAF electronics” and added to the “Raspberry Pi and Servo Hat” section. We increased the picture sizes and added more legends to the ZAF+- Hardware “Pumps and Valve''. Moreover, we added more photos to the “tubing” section and the “ZAF+ Electronics” section. We agree that videos or gifs would have been great to visualize the assembly. Unfortunately, we did not record such videos during the construction. We created ZAF as an open source project and clearly hope to generate a community that will share assembly pro-tips and may be constructions videos on the GitHub.

Our institute is expanding on zebrafish research so we will build additional ZAFs and will use this opportunity to prepare nice videos to add to the wiki. We envision that the wiki will be improved over time with better material, some of it contributed, as well as perhaps newer and better versions of ZAF.

The main question that would affect if this approach were taken up would be how reliable it is in the long run. Have the authors experienced any issue over the 2 months test? Is this system still being used currently? If so, could the authors update the water quality logs?

The reviewer suggests that the key question is to see if using ZAFs all year long is possible. We can reply yes, it is actually possible! We have used ZAF for 9 months, and now ZAF+ for the past 10 months in our fish facility, with great success. We never experienced major malfunctions and the minor issues we encountered are reported in the troubleshooting table. Since ZAF and ZAF+ have been used daily for months with logs recorded every day we have updated the water logs quality to 3 months. We have been using the ZAFs in full autonomy for a total of 19 months, frankly invaluable.

Getting a sense of how long it can run without problems, how much troubleshooting is involved per month would be very useful in answering those questions.

Except manual cleaning and tube replacement, there is no other big maintenance on ZAF. Of course, the food reserve needs to be changed at least once per week. We listed the malfunctions in the troubleshooting guide in the wiki. In our facility ZAFs require an average of 1 hour of maintenance per month. And if any hardware part fails you can just immediately replace it because all the parts are cheap and easily replaceable. Actually, we recommend keeping spare parts of all the key components (pumps, valves, Arduino, Raspberry Pi, tubes,.…).

Minor commentsMain text page 3: Figure Supp. 2 instead of Supp. Figure 2. Furthermore, would the authors have similar data for the manual feeding? If so, it could be useful to add here for comparison (although that is not necessary if the data is unavailable).

We changed the text, but we don’t have data available for the water logs with manual feeding.

Main text page 3: it would be useful to add how long it takes to change all the tubing after 10 weeks?

This is really dependent on ZAF tubing and the fish facility, in our hand for about one hour. We mentioned it in the Results section, ZAF paragraph.

Methods fish weight: The phrasing as it stands make it unclear the same method was used for ZAF and ZAF+, the authors may consider starting with the description of the common weighting method, then the specifics of ZAF+.

Thank you, we changed the text accordingly.

Supp. Figure 1a: "Waste water drain pipe"

Thank you, we changed the text accordingly.

Acknowledgments: "…for their help…"

Thank you, we changed the text accordingly.

ZAF – Servo Hat connection: "to control the pumps"

Thank you, we changed the text accordingly.

ZAF – Installation: the dependencies should be listed as they are in ZAF+, or the two sections merged, unless the GUI is not functional (see below).

Thank you, we now list the dependencies in the wiki.

ZAF – How to use: there is no mention of the GUI, is it not yet implemented? If not, is the touch screen needed?

The standard ZAF hardware is controlled by a very simple python-based program that works with a command line interface. Therefore to interact with the Raspberry Pi for installation and configuration we strongly recommend building ZAF with a screen, and the touch screen is an easy way to be able to quickly point and click in the absence of a mouse – which can be cumbersome when no clean horizontal surfaces are available in a lab environment.

ZAF+ – soldering: "A 12V power supply (at least 10A best 20A) provides power to the electronics, except the RaspberryPi and the two Arduino Megas." It seems the sentence is incomplete, or at least I cannot make sense of it.

Changed to “A 12V power supply (at least 10A, but ideally 20A) provides power to the electronics, except for the RaspberryPi and the two Arduino Megas that are powered by the Raspberry Pi 5V GPIOs.”

Reviewer #2 (Significance (Required)):This manuscript provides a significant technical advance to the zebrafish field. The proposed automated feeder would be a very useful option for smaller labs, to ensure the consistency of feeding, and to remove one of the routine aspects of fish husbandry.As the authors state, there is certainly interest in the zebrafish community [9,10] for automation of feeding. I am not aware of other DIY fully automated feeding system, commercial systems do exist, but are expensive.The manuscript, and proposed automated feeder, would certainly be of interest within the zebrafish community, as well as other researchers using aquatic models that can rely on dry food. How many in the community would embrace this method will depend on how confident they are in the long-term stability.I am neither electronics, nor husbandry expert. As such I am not qualified to comment on any long-term approach this may prove, if any, for fish health. My expertise lies in image and data analysis, as well as microscopy.Referee Cross-commentingI think the major points are shared by all reviewers, I think the other reviews are fair in their content and I have nothing specific to comment on.Reviewer #3 (Evidence, reproducibility and clarity (Required)):SummaryThis technical report describes an open-source fully automated feeding system for husbandry of zebrafish (and potentially other aquatic organisms). It provides detailed instructions for assembling individual components into two different feeding systems of varying adaptability, as well as their operation. Links to relevant control software are also provided. The characterization of the systems' performance appears somewhat limited (e.g. only maintenance of adult fish over a period of 8 weeks and use of dry food is documented). These systems could be of use for husbandry in a large number of research labs, and, in addition, for automated reward delivery in large-scale associative conditioning assays.

We thank the Reviewer for his encouraging comments and appreciate his helpful suggestions. We answer to the Reviewer comments bellow:

Major commentsProviding food to large numbers of tanks in aquatic animal facilities in a regular fashion is a time- and resource-consuming process. Some automated feeding systems for large numbers of tanks are commercially available, but these feeder robots are expensive and are restricted to systems of specific vendors. Therefore, an adaptable automated system that can be assembled from off-the-shelf components is a very attractive option for many research labs to both save resources and standardize the feeding process.The instructions for assembly provided by the authors appear quite detailed and sufficient to allow non-experts the assembly and operation of the automated feeder systems. The design of the system appears appropriate for the task.While additional experiments are not required to support the claims of the article, I feel that it would be significantly improved by the provision of additional information. My suggestions in that regard include:Description of the washing procedure of the system (which solvents, how often, how long?). The authors mention that an exchange of the tubing is required every 10 weeks, but since the tubing transports liquid food mixture, it is easily conceivable that microbial growth will occur rapidly in the system without thorough hygiene / washing procedures. Also, could the authors provide some information, which type of tubing material they are using (Silicone, Tygon etc.)?Description of the washing procedure of the system (which solvents, how often, how long?).

We agree that the cleaning procedure must be clarified. Therefore, we added a more clear description of the process in the first paragraph of the discussion and clarified the explanation about cleaning in Box 1 and Box 2 (suggested also by the reviewer1). To summarize there are two levels of cleaning, the first one happens just after a food distribution program by flushing water and air in the system (Box1). Additionally, at least once a day, we run an entire program without food, to rinse/clean the system (Box2). This last step is programmable using ZAFs software.

The authors mention that an exchange of the tubing is required every 10 weeks, but since the tubing transports liquid food mixture, it is easily conceivable that microbial growth will occur rapidly in the system without thorough hygiene / washing procedures

Following all reviewers' comments, we added an extra supplementary figure justifying the need of changing the tubes every 12 weeks (updated based on our latest observations). We monitored the cleanliness (algal/microbial growth) of the tubes and realized that it becomes necessary to replace the tubes every 12 weeks (supp figure 3). Interestingly, we remarked that the microbial and algal growth depends on the facility specificities such as light intensity and temperature.

Also, could the authors provide some information, which type of tubing material they are using (Silicone, Tygon etc.)?

For ZAF we used silicone-based tubing then we changed to PVC based tubes for ZAF+ because they are cost effective and have similar specifications for our usage. We added a note about the tubing material in the wiki ZAF tubing and ZAF+ tubing.

In a related point, I was left wondering how long the food is being mixed in the mixing flask before being applied to the animals? Too long mixing might lead to a loss of nutrients into the solution (through diffusion).

Very relevant point, indeed it is very important for the food to not be mixed too long in water to avoid pellet dissolution in water and loss of nutrients. The food manufacturer website mentioned: “duration of “wet” feeding should be kept short”

(https://zebrafish.skrettingusa.com/pages/faq). Therefore, we adapted our feeding program to keep the “wet” feeding extremely short. For ZAF and ZAF+, the software is designed to deliver the mix of food and water to tank(s) within 3 minutes at most. To clarify this, we added in the Box describing the feeding, a sentence: “Overall, they share many common features, like the quick distribution of food and water mix, to avoid pellet dissolution in water and loss of nutrients.”

Do the food pellets remain more or less integral so that the majority of delivered food is actually ingested by the fish?

We manually evaluated the integrity of food pellets in the early phase of development, these parameters being difficult to quantify, we decided to record the fish weight as a readout of good food delivery and general effectiveness. However, we clearly understand the reviewer's remarks and therefore added to the manuscript a supplementary video that shows the distribution of the food pellets and their integrity once they reach the tanks.

In yet another related point, I was left wondering, whether the authors observed any negative impact of feeder usage on water quality (besides pH and conductivity, which they report)? Especially, with regards to ammonia that might arise from the decomposition of uneaten food items?

Ammonia toxicity is mentioned to induce clinical and microscopic changes that reduce growth and increase susceptibility to pathogens according to aquaculture textbooks as summarized here:

(https://zebrafish.org/wiki/health/disease_manual/water_quality_problems#ammonia_toxicity). However, we never experienced such abnormal phenotypes in our facility and our regular aquatic PCR health monitoring profiles have always been negative for pathogens. Additionally, high ammonia is influenced by husbandry conditions, such as important fish density or inappropriate water circulation, characteristics that are not present in our fish facility. Therefore, we did not find relevant to test for ammonia levels.

The authors only tested the feeder on adult fish but discuss that it would easily be transferable to a system that is used for raising fish fry. In that context, could the authors comment, on whether the system of using water as the carrier for the dry food (after mixing) would work as well for the smaller pellets required in feeding fish fry (e.g. 75 or 100 μm pellet size as compared to the 500 μm pellet size they use)? With smaller pellets, break-down of the dry food during the mixing process seems to be an even larger problem, I could imagine.

We appreciate the reviewer's comment about using different food pellets sizes, a very important point for ZAFs adoption beyond adult fish. During ZAFs testing we actually tested different food sizes (from 100uM pellets to 500uM) and did not observe differences in pellet distribution. Most of the industrial aquatic food pellets are oily and designed for automatic distribution (for large farming environments). Therefore, they keep their integrity and are not easily broken. Besides, during food distribution, as mentioned previously, the duration of wet food (water and food mix) is relatively short, which helps maintain pellet integrity.

Minor comments1. The average weight of animals is given as lying in the range of 5 to 6g. That seems very high. The "standard" weight range of adult zebrafish is more around 1g [see, for example: Clark, T. S., Pandolfo, L. M., Marshall, C. M., Mitra, A. K. and Schech, J. M. Body Condition Scoring for Adult Zebrafish (*Danio rerio*). j am assoc lab anim sci (2018)]. Could the authors comment on that discrepancy?

Good observation by the reviewer. We did make a mistake during figure preparation and our legends were actually not reflecting the exact weight of the fish. The scale bars of the figures have been changed to reflect the real weight of the fish (below 1g). We thank the reviewer for noticing the mistakes.

2. The authors state that spawning success is not negatively affected by the automated feeding, and they quantify the number of successful crosses. Could the authors briefly confirm or state, that or whether the clutch size was also unaffected?

We never precisely quantified the clutch size nor quality, but we are now using ZAFs for the feeding of our facility for 19months and never observed any problem with our clutch. Our lab is working on early development and crucially relies on clutch quality.

3. The manual feeding procedure / regime that is used to compare husbandry success against the automated feeding regime is not described in any detail. That seems important given the topic of the article.

We agreed and added a brief description of the protocol in the Methods section (“Animal and husbandry”).

4. The authors cite two recent papers that describe semi-automatic feeding systems for zebrafish in the introduction. The authors might want to consider discussing some key differences between their system and these semi-automatic systems in the discussion.

The two published semi-automatic feeding systems are completely different from the devices presented in our paper. They are also open access, but they are devices that need to be manually operated by facility staff. In contrast, our solutions are fully automatic and do not require the human hand during operation. We mention these two solutions during our brief literature overview in the introduction. However, since these are in a different category, we did not judge it necessary to comment on them in the discussion.

5. What do the error bars in Figure 1c signify (s.d., s.e.m.)? Please state in Figure legend.

We thank the reviewer for their attention to details and explain in the figure that we mean standard error of the mean by s.e.m.

6. I do think that the system could be of particular interest to researchers that study learning and that use food rewards in automated associative conditioning experiments. While this might be obvious to researchers with such an interest, this aspect is not at all discussed in the paper. Mentioning it might further underscore the versatility of the feeder system.

We agree with the reviewer that ZAF can be adapted to experimental conditions such as behavioral conditioning, nutrition and drug delivery. Any experiment requiring the automatic delivery of solid pellets or liquid can benefit from ZAF. We revised our text and mentioned it in the discussion.

7. A list of all required equipment with vendors and price estimates (e.g. in the Supplement) would make this paper an even more readily accessible resource.

This is a very important point already suggested by another reviewer. We added two extra tables in the wiki with the necessary parts and components, listing models, references, and prices.

Reviewer #3 (Significance (Required)):Describe the nature and significance of the advance (e.g. conceptual, technical, clinical) for the field.This article signifies a purely technical advance in that it provides a characterization of an opensource, scalable automated feeder for aquatic facilities. As such, it presents a significant advance in the field of aquatic animal husbandry. In addition, this system could also be useful for automated large- or medium-scale associative conditioning paradigms, in which food rewards are given as positive reinforcers.Place the work in the context of the existing literature (provide references, where appropriate). The authors refer to previously published semi-automatic feeder systems. Regardless of the advantages or disadvantages of all these systems, the field will benefit from a broad(er) choice of automatic feeding systems that are described in sufficient detail to be easily assembled in the laboratory.State what audience might be interested in and influenced by the reported findings. This study is of interest for any research laboratory working with zebrafish or other aquatic model organisms. Thus, the audience for this article is very broad. Specific interest might also arise in researchers that are performing learning studies in zebrafish (see above).Define your field of expertise with a few keywords to help the authors contextualize your point of view. Indicate if there are any parts of the paper that you do not have sufficient expertise to evaluate.Zebrafish, neural circuits, sensory systems.Referee Cross-commentingMany of the major points are shared by all three reviewers. Beyond these shared points, I agree.